# Inhibition of MCP1 (CCL2) Enhances Antitumor Activity of NK Cells Against HCC Cells Under Hypoxia

**DOI:** 10.3390/ijms26104900

**Published:** 2025-05-20

**Authors:** Hwan Hee Lee, Juhui Kim, Eunbi Park, Hyojeung Kang, Hyosun Cho

**Affiliations:** 1College of Pharmacy, Duksung Women’s University, Seoul 01369, Republic of Korea; oeo3oeo@gmail.com (H.H.L.); kimjuhui@duksung.ac.kr (J.K.); peb8956@duksung.ac.kr (E.P.); 2Duksung Innovative Drug Center, Duksung Women’s University, Seoul 01369, Republic of Korea; 3Vessel-Organ Interaction Research Center, VOICE (MRC), Cancer Research Institute, College of Pharmacy, Kyungpook National University, Daegu 41566, Republic of Korea

**Keywords:** MCP1 (CCL2), hypoxia, natural killer (NK) cell, tumor microenvironment, human hepatocellular carcinoma (HCC)

## Abstract

Hypoxia, a low-oxygen state, is a common feature of solid tumors. MCP1 (CCL2) is a small cytokine that is closely related to hypoxia and has a positive effect on tumor development. Hypoxia causes resistance to various treatments for solid tumors and the evasion of cancer immune surveillance by lymphocytes. Natural killer (NK) cells are innate lymphocytes that play an important role in cancer development, particularly in the liver. First, it was found that the incubation of HCC in hypoxia (2–5% O_2_) significantly increased the production of several inflammatory cytokines, including MCP1, compared to that of normal oxygen (20% O_2_). Subsequently, blocking MCP1 with an anti-MCP1 antibody in HCC cultures inhibited the growth and migration of HCC cells in vitro and in vivo. This was associated with a decrease in the expression of HIF-1α/STAT3 in HCC under hypoxia. Furthermore, blocking MCP1 in HCC cell cultures under hypoxia significantly increased the chemotaxis and activation of NK-92 cells against HCC cells. MCP1 blockade in HCC cell cultures under hypoxia induced a shift in NK cells to the CD56^+dim^ population and an increase in the expression of the activation receptors NKG2D and NKp44. In conclusion, modulation of MCP1 could enhance NK activity against hypoxic HCC cells.

## 1. Introduction

The tumor microenvironment (TME) is composed of various factors that interfere with the immune cell surveillance of cancer cells [1]. Hypoxia, a low-oxygen state, is a hallmark of the TME present in many solid tumors and causes resistance to various cancer treatments, including radiotherapy, chemotherapy, and immunotherapy [2]. Inflammation plays an important role in disease progression [3] and is mediated by pro-inflammatory cytokines produced by various cells, including immune cells, epithelial cells, and endothelial cells [4]. The uncontrolled inflammatory state that occurs when pro-inflammatory cytokine levels are high and persistent has been shown to be closely associated with advanced cancer [5,6]. Recent studies have shown that the hypoxic environment in tumors increases the levels of several pro-inflammatory cytokines, interleukin (IL)-6, tumor necrosis factor-alpha (TNF-α), and interleukin (IL)-1 [7,8,9]. Previous studies have shown that high levels of IL-6 in the TME of highly invasive breast cancer and hepatocellular carcinoma (HCC) have a positive effect on cancer cell growth and metastasis, as well as on the attenuation of natural killer (NK) cell immune surveillance [10,11,12]. IL-6 is known to induce the expression of several chemokines, including monocyte chemoattractant protein (MCP)-1, also known as CCL2 [13]. MCP1 is a chemokine, which is a small cytokine (chemotactic substance) that attracts immune cells, such as neutrophils and macrophages, to sites of inflammation or injury [14]. However, recent studies have shown that MCP1 is involved in the development and metastasis of several types of cancer, including breast, prostate, and pancreatic cancer [15,16,17]. In addition, MCP1 induces hypoxia in the TME by recruiting inflammatory cells.

Hypoxia-inducible factors (HIFs) are transcription factors that are activated by levels of hypoxia and help cells adapt to hypoxia by regulating cellular metabolism [18]. The activation of HIFs plays an important role in cancer progression, including angiogenesis and metastasis [2]. HIFs consist of the HIF-α (HIF1α or HIF2α) subunit, which is only expressed under hypoxic conditions, and the HIF1β subunit, which is always expressed [19]. Therefore, HIF-1α expression has been observed in primary and metastatic breast cancer but not in normal tissue around the cancer [20]. In addition, some studies have shown a high correlation between HIF-1α expression and advanced cancer, and it was observed that the higher the HIF-1α expression is, the higher the tumor stage is [20,21]. HIF-1α activation is regulated oxygen-dependently by the von Hippel–Lindau tumor suppressor protein (pVHL)-dependent pathway [22] and oxygen-independently by growth factors, cytokines, and other signaling molecules. The activation of phosphatidylinositol 4,5-bisphosphate 3-kinase (PI3K), a protein involved in cytokine signaling, allows the upregulation of HIF-1α protein translation [23], and the MAPK pathway also regulates HIF-1α synthesis and transcriptional activation [24]. Recent studies have shown that activation of signal transducer and activator of transcription 3 (STAT3) and HIF-1α in innate immune B cells stimulated by lipopolysaccharide (LPS) can exacerbate cancer metastasis [25]. STAT3 regulates the production of cytokines such as IL-6, a signaling molecule involved in immune response and inflammation. MCP1 also induces activation of the JAK/STAT3 and PI3K/AKT pathways in cells [26], which contribute to various physiological functions.

Natural killer cells (NK cells), which are innate immune lymphocytes, play an important role in tumorigenesis [27]. Since NK cells account for approximately 25–50% of liver lymphocytes, NK cells are intimately involved in the development of liver cancer. The number of NK cells in the blood and tumor tissues of patients with HCC is closely related to survival and prognosis [28]. Chemokines secreted by various cells contribute to disease progression by recruiting immune cells in the early stages of disease. Therefore, some chemokines present in the TME may exert antitumor activity by recruiting natural killer cells to the tumor site [29]. However, recent studies have shown that inflammatory chemokines such as CXCL1, CXCL2, and CXCL8 present in the TME can negatively affect NK cells, leading to NK cell depletion or dysfunction [30,31,32]. In addition, a more recent study showed that blocking the MCP1 (CCL2)-CCR2 interaction with a CCL2-neutralizing antibody in a murine hepatocellular carcinoma model reduced the number of inflammatory myeloid cells (CD11^highGr1+)^ and increased the cytotoxicity of hepatic NK cells [33]. First, it was confirmed that the level of MCP1 was increased in the hypoxic HCC cell culture. However, the relationship between MCP1 and hypoxic HCC and the effect of MCP1 on the activity of NK cells against cancer have not been clearly elucidated. This study showed that high levels of MCP1 in a hypoxic HCC cell culture were blocked by anti-MCP1 antibodies, demonstrating a direct effect on cancer cells and an effect on the activity of NK cells.

## 2. Results

### 2.1. Increased Production of MCP1 by HCC Cells Under Hypoxia

The differences in cytokines produced by HCC cells under normoxia or hypoxia were analyzed using multiple cytokine arrays. Several cytokines, including GRO-α (CXCL1), IL-6, MCP1 (CCL2), and angiogenin, were produced more in hypoxic HCC Luc-SK-Hep1 cells than in cells under normoxic conditions (Figure 1A). Among them, the production of the pro-inflammatory cytokines MCP1 and IL-6 was significantly increased in cells under hypoxic conditions (Figure 1A–C; * *p* < 0.05, *** *p* < 0.0001). The high production of IL-6 in HCC cells has been identified in previous studies [12], which was investigated in HCC cells artificially expressing HIF-1α by CoCl_2_.

### 2.2. The Inhibition of HCC Cell Growth by the Reduced Expression of HIF-1α Through the Blockade of MCP1 in the Presence of Hypoxia

Hypoxic-inducible factor (HIF)-1α was significantly expressed in HCC Luc-SK-Hep1 cells at low oxygen (2% O_2_) but not at normal oxygen (20% O_2_), as shown in Figure 2A. Treatment with antibodies (human anti-IL-6; human anti-MCP1) in the cell culture under hypoxia resulted in a significant inhibition of HIF-1α expression in the cells (Figure 2A; * *p* < 0.05; *** *p* < 0.0001). It also resulted in the decreased phosphorylation of STAT3 in the cells (Figure 2B). In addition, cell viability was significantly reduced upon treatment with anti-IL-6 or anti-MCP1 (Figure 2C; *** *p* < 0.0001), while cell apoptosis was increased (Figure 2D; ** *p* < 0.001).

### 2.3. Inhibition of HCC Cell Migration by Blockade of MCP1 Under Hypoxia

Cellular mobility is important in tumorigenesis because its ability affects the invasion and metastasis of cancer cells. The mobility of HCC Luc-SK-Hep1 cells was determined by wound healing assay and transwell assay. In the wound healing assay, the migration of Luc-SK-Hep1 cells was decreased by treatment with anti-IL-6 or anti-MCP1 compared to untreated cells (Figure 2E; * *p* < 0.05) when it was incubated for 24–48 h under hypoxia. The blockade of IL-6 for 24 h in cell culture medium under hypoxia did not result in a difference in cell spreading compared to untreated, but the blockade of MCP1 in cell culture medium was different when incubated for both 24 and 48 h. Cell migration using the transwell assay showed that treatment with anti-IL-6 or anti-MCP1 for 24 h significantly reduced the number of cells in the transwell membrane compared to no treatment (Figure 2F; * *p* < 0.05; ** *p* < 0.001).

### 2.4. Suppression of HCC Tumorigenesis by Anti-CCL2 (MCP1) Treatment in Mice

To analyze the anti-tumorigenesis of HCC by anti-MCP1 treatment in vivo, SCID mice were used. Hypoxia in the TME is a common feature of several solid tumors [34,35,36]. Therefore, we investigated whether the knockdown of the HIF-1α gene in HCC Luc-SK-Hep1 cells affects their tumorigenesis. The knockdown of the HIF-1α gene in Luc-SK-Hep1 cells inhibited the growth of tumors generated by the subcutaneous injection of cells into mice (Appendix A). Subsequently, the anti-tumorigenic effect was verified in mice by anti-MCP1 treatment after tumor development by the intravenous injection of Luc-SK-Hep1 cells. Antibody treatment by intraperitoneal injection suppressed tumor metastasis and had a significant effect at 4 weeks compared to in untreated groups (Figure 3C,D; * *p* < 0.05; ** *p* < 0.001). In addition, the expression of HIF-1α and pSTAT3 was significantly decreased in antibody-treated tumors compared to in untreated tumors (Figure 4A,C; * *p* < 0.05; ** *p* < 0.001). The expression of N-cadherin and MMP9, which are related to tumor metastasis, was also significantly decreased in tumors from mice in the antibody treatment group compared to those in the untreated group (Figure 4B,C; * *p* < 0.05; ** *p* < 0.001).

### 2.5. Improvement of NK Activity Against HCC ceLLs by Blockade of MCP1 Under Hypoxia

A previous study showed that the expression of HIF-1α in HCC cells decreased the NK activity, which was associated with high levels of IL-6 [12]. Here, NK cytotoxicity against HCC cells was attenuated under hypoxia compared to normoxia (Figure 5A; 20% O_2_, 22.14%; 5% O_2_, 10.57%; *** *p* < 0.0001), and the expression of the activating receptors NKG2D and NKp44 on the surface of NK-92 cells was also significantly reduced (Figure 5E; NKG2D^+^: 20% O_2_, 82.5%; 5% O_2_, 76.01%; NKp44^+^: 20% O_2_, 36.71%; 5% O_2_, 32.34%; * *p* < 0.05; *** *p* < 0.0001). In addition, the mobility of NK-92 cells into HCC cells was decreased under hypoxia compared to normoxia (Figure 5C). However, this effect was attenuated by treatment with IL-6 or MCP1 antibodies under hypoxia. NK cytotoxicity against HCC cells was significantly improved by the blockade of IL-6 or MCP1 in the conditioned media of their co-culture (Figure 6F; untreated 8.00%, anti-IL-6 14.20%, anti-MCP1 18.24%, Comb 17.41%; * *p* < 0.05; ** *p* < 0.001). When co-cultured with HCC cells under hypoxia, CD56^+dim^ NK cells were also significantly increased, and the expression of NK-activating receptors NKG2D and NKp44 on the surface of NK cells was enhanced (Figure 6A–D; CD56^+dim^: untreated 8.59%, anti-IL-6 10.77%, anti-MCP1 13.06%, Comb 15.80%; NKG2D^+^: untreated 47.47%, anti-IL-6 53.96%, anti-MCP1 54.64%, Comb 54.34%; NKp44^+^: untreated 8.69%, anti-IL-6 11.30%, anti-MCP1 11.95%, Comb 17.66%; * *p* < 0.05; ** *p* < 0.001; *** *p* < 0.0001). However, when NK-92 cells were co-cultured with MCP1 shRNA-transfected HCC Luc-SK-Hep1 cells, only NKp44 was significantly expressed on the surface of NK cells compared to the co-culture with shcontrol under hypoxia (Figure 6G; shcontrol 51.64%, shMCP1 67.60%; * *p* < 0.05). The apoptosis of target cells by cytotoxic T lymphocytes (CTLs) and natural killer cells is triggered by two representative signals: the induction of Fas (CD95) on the surface of target cells or release of granzyme B or perforin into target cells [37]. The level of Fas expressed in HCC cells was increased in the co-culture with MCP1 shRNA-transfected HCC cells compared to that with shcontrol (Figure 6H,I).

## 3. Discussion

In a previous study, it was confirmed that the activity of NK cells in HCC expressing HIF-1α by treatment with CoCl_2_ was increased due to the blocking of IL-6 in the tumor microenvironment (TME) [12]. However, the effects of high levels of inflammatory cytokines on the anti-tumor activity of NK cells in HCC cultures under hypoxic conditions have not been elucidated. Pro-inflammatory cytokines present in the TME have been controversial with regard to their antitumorigenic or protumorigenic activity. However, recent studies have shown that pro-inflammatory cytokines have a greater effect in causing protumorigenesis. The pro-inflammatory cytokine IL-6 is secreted more in invasive breast cancer (triple-negative breast cancer; TNBC) than in non-invasive breast cancer and affects tumor development and metastasis [10]. In other words, IL-6 has been shown to confer motility and invasiveness to non-invasive cancer cells. High levels of IL-6 in the TME promote tumor cell proliferation, survival, and invasiveness through the IL-6/JAK/STAT3 signaling pathway [38]. STAT3 is activated by inducing NF-kB transcription through signals associated with various cytokines and growth factors, including IL-6 [39]. Recent studies have shown that MCP1 signaling can promote cancer cell migration and invasion through the activation of STAT3 in various cancer cells [14,40]. MCP1 is a chemokine (small cytokine) that can attract immune cells to lesions, but in the TME, it is an inflammatory mediator involved in tumor development [41]. In hypoxia (2–5% O_2_), the level of IL-6 and MCP1 were increased in invasive HCC (Figure 1). The relationship between hypoxia and inflammation has been well established by many studies [42]. For example, hypoxia is a common feature of inflammatory bowel disease (IBD), and the inflammatory process and HIF-1 activation are linked [43]. In other cases, hypoxia is caused by high oxygen consumption due to the high metabolic demands of cancer, and this leads to the induction of inflammation [18].

Cancer cells transcribe the HIF-1α gene to adapt to the low-oxygen conditions of hypoxia [44]. HIF-1α is overexpressed in solid tumors compared to normal tissue [45], indicating poor prognosis in patients with HCC [46]. In xenograft model experiments, mouse tumors generated by the subcutaneous injection of HCC cells did not grow well in HIF-1α shRNA-transfected HCC compared to controls (Appendix A). The positive correlation between IL-6 and HIF-1α has been reported in several studies. Blocking the interaction of IL-6 and IL-6R in colitis-associated colorectal cancer (CAC) by anti-IL-6 receptor antibodies reduced the size and number of tumors in mice, and the effect was mediated by the downregulation of HIF-1α [47]. A previous study also showed that the blockade of IL-6 with anti-IL-6 antibodies in HCC cell culture reduced HIF-1α expression [12]. IL-6 strongly induces the mRNA expression and secretion of MCP1 in peripheral blood mononuclear cells (PBMC) and the human myeloid cell line U937 [13]. However, in this study, IL-6 levels in the culture medium were reduced in HCC cell cultures transfected with MCP1 shRNA and in HCC cell cultures treated with anti-MCP1 antibody, whereas MCP1 levels were not reduced in HCC cell cultures treated with anti-IL-6 antibody (Appendix A). More recent studies have shown, by sequence analysis, that the HIF-1α binding site is located in the promoters of MCP1 and MCP5. In vitro and in vivo experiments have also confirmed that the downregulation of IL-6 or MCP1 (CCL2) in hypoxia inhibits HIF-1α expression in hepatocellular carcinoma. Hypoxia triggers the epithelial-to-mesenchymal transition (EMT) process, which affects the motility of cancer cells. HIF-1α can interact with regulatory factors such as snail, slug, twist, and β-catenin to promote the EMT process, resulting in increased expression of N-cadherin and vimentin [48]. The overexpression of N-cadherin is associated with increased cancer cell invasion and metastasis [49]. The knockdown of MCP1 in the TNBC cell line BT-549 reduced cell invasion through the downregulation of N-cadherin and vimentin [15]. Tumor metastasis in mice derived from the intra-tail vein injection of HCC cells was significantly inhibited compared to that in untreated mice when intraperitoneally injected with anti-IL-6 or anti-MCP1 antibodies, due to the inhibitory effect of N-cadherin expression in tumors in mouse liver tissue (Figure 3 and Figure 4).

NK cells are innate lymphocytes that are very important for tumor development and have cytotoxic activity that directly kills cancer cells. Several studies have shown that the frequency of tumor-infiltrating NK cells is associated with patient mortality in various cancers [50]. Hypoxia, one of the characteristics of the TME, alters the biological activity of lymphocytes and weakens their anti-cancer function. In hypoxic tumors, effector T lymphocytes are not only reduced in their migration from the circulation into the tumor but also weaken in their anti-tumor activity [51,52]; NK activity against cancer cells is also low due to the decreased expression of several activating receptors, including NKG2D, NKp30, and NKp46 [53]. In addition, HIF-1α transcription in the melanoma cell line B16-F10 results in an increase in tumor-infiltrating NK and CD8^+^ T cells [54]. As mentioned above, the increased expression of HIF-1α in HCC increases the production of inflammatory cytokines such as IL-6 and MCP1. The release of MCP1 by IL-6 signaling in human vascular endothelial cells induces the activation of the JAK/STAT3 and PI3K/AKT pathways [26], also in various cancer cells [38]. Several studies have shown that the activation of STAT3 in tumor cells attenuates tumor surveillance by NK cells through cytokine secretion [55,56]. MCP1 is primarily known to be involved in the recruitment of NK cells to tumors as an inflammatory mediator [57], but more recently, the tumor expression of MCP1 can inhibit the function of NK cells against tumors [58]. In this study, blockade with anti-MCP1 and anti-IL-6 antibodies alone or in combination under hypoxia increased NK cell killing activity against HCC through the upregulation of NKG2D and NKp44, which are NK activation receptors (Figure 6). In addition, interestingly, NKG2D expression was not increased by combination therapy, but NKp44 expression was increased. Several studies have reported that the IL-6/JAK/STAT3 pathway is associated with the suppression of NKG2D expression [59,60]. However, the effect of CCL2 on the NK-activating receptor has not been clearly reported. As shown in Figure 2A, blocking IL-6 and MCP1 inhibited STAT3 and HIF-1α expression, which likely induced NK cell receptor activation. However, since IL-6 and CCL2 have different receptors, the signaling pathways associated with the differences in ligand expression on the active receptors on the cancer cell surface may have been different. In addition, blocking MCP1 significantly increased the chemotactic activity of NK cells against hypoxic HCC compared to the control group. Previous studies have shown that blocking IL-6 increased anti-cancer NK cell activity in HCC cell cultures induced by CoCl_2_ [12]. However, it appears that the cytokine that directly affects HIF-1α expression under hypoxia is the regulation of MCP1. Therefore, these results suggest that neutralizing the local cytokine MCP1 to inhibit HIF-1α may overcome hypoxia-induced resistance in solid tumors and enhance the antitumor activity of NK cells.

## 4. Materials and Methods

### 4.1. Cell Maintenance

The human HCC cell line SK-Hep1 with luciferase insertion was obtained from Professor Kuroda (Osaka University) and maintained in Minimum Essential Media (GenDEPOT, Katy, TX, USA) supplemented with 10% fetal bovine serum (FBS; Gibco, Grand Island, NY, USA), 100 U/mL penicillin, and streptomycin (Gibco, Grand Island, NY, USA). The NK cell line NK-92 was provided by American Type Culture Collection (ATCC, Manassas, VA, USA). The cells were cultured in alpha-MEM (Gibco) with the addition of 20% FBS, 0.1 mM 2-mercaptoethanol (Sigma Aldrich, St. Louis, MO, USA), 100 U/mL penicillin and streptomycin (Gibco), and 200 U/mL of rhIL-2 (BioLegend, San Diego, CA, USA). All cells cultured in complete medium were maintained at 37 °C in a humidified atmosphere containing 5% CO_2_. The hypoxic condition was 5% oxygen for most experiments, but 2% oxygen was used for the protein expression experiment. To induce a hypoxic environment, we used the oxygen concentration settings mentioned in previous studies [61].

### 4.2. Gene Knockdown Using shRNA Plasmid Transfection

Both the HIF-1α shRNA plasmid and the MCP1 shRNA plasmid were purchased from Santa Cruz Biotechnology Inc. (Dallas, TX, USA). To edit the genes in Luc-SK-Hep1 cells, the cells were incubated overnight at a density of 1 × 10^6^ cells per each 6-well plate to more than 90% cell confluency. The cells were then added to the mixture diluted at the appropriate ratio of shRNA plasmid and shRNA plasmid transfection reagent (Santa Cruz Biotech., Dallas, TX, USA) in shRNA plasmid transfection medium (Santa Cruz Biotech.) and incubated for 5 h. After incubation, the cells were incubated for 24 h at 1× serum of a final concentration in medium supplemented with 20% serum. To select only cells transfected with shRNA plasmid, the cells were incubated in growth medium supplemented with puromycin.

### 4.3. Cytokine Array

Cytokines produced by the effector (NK cell) and target (Luc-SK-Hep1 cell) co-culture or alone were analyzed using the Human cytokine array C3 (RayBiotech, Norcross, GA, USA), Human IL-6 ELISA Set (BD Biosciences, Franklin Lakes, NJ, USA) and BD OptEIA™ Human MCP1 ELISA set (BD Biosciences). Briefly, target cells were plated overnight at a density of 5 × 10^5^ cells per each in 6-well plate in a condition of varying oxygen concentration (normoxia: 20%; hypoxia: 2–5%) and then co-cultured with effector cells at a ratio of 1:2 (T:E) for 4 h. After the co-culture, the supernatants were collected and centrifuged at 2000 rpm at 4 °C for 10 min. The supernatants were used for cytokine analysis. All cytokine analysis experiments were performed according to the manufacturer’s instructions.

### 4.4. Cell Apoptosis Using Annexin V Staining

BD Pharmigne™ APC Annexin V (BD Biosciences) was used to analyze cell apoptosis after antibody treatment. Briefly, cells from the HCC cell line Luc-SK-Hep1 were cultured at a density of 5 × 10^5^ cells per each 6-well plate overnight and then treated with 2 ng/mL of purified anti-human IL-6 antibody (BioLegend, San Diego, CA, USA) or 1 ng/mL of purified anti-human MCP1 (CCL2) antibody (BioLegend) for 24 h to block the interaction. After the treatment, the cells were harvested and stained with APC annexin V for 15 min. Cells were assessed using flow cytometry (Novocyte Flow Cytometer, ACEA Biosciences, San Diego, CA, USA). Apoptotic cells were analyzed in comparison to unstained cells.

### 4.5. Western Blot Analysis

Briefly, protein samples were extracted from cells or tissues using protein extraction buffer (Intron, Seoul, South Korea) and quantified using the Bradford assay (Coomassie blue, GenDEPOT). The quantified protein was separated by electrophoresis, transferred to a microporous polyvinylidene fluoride membrane (PVDF, Millipore, Burlington, MA, USA), and blotted with the first and second antibodies. Membranes were immersed in enhanced chemiluminescence (ECL) reagent and detected using Chemi-doc (Millipore) or ImageQuant™ (GE Healthcare Life Sciences, Marlborough, MA, USA) LAS 500. Primary antibodies: HIF-1α (Novus Biologicals, Centennial, CO, USA), (p)-STAT3 (Cell signaling, Danvers, MA, USA), MMP9 (Cell signaling), N-cadherin (Cell signaling), Granzyme B (Santa Cruz Biotech.), Fas (Santa Cruz Biotech.), and β-actin (Sigma-Aldrich, St. Louis, MO, USA). Secondary antibodies: goat anti-mouse IgG-HRP (Santa Cruz Biotech.) and mouse anti-rabbit IgG-HRP (Santa Cruz Biotech.).

### 4.6. Cell Migration Assay

Luc-SK-Hep1 cells were plated at a density of 5 × 10^4^ cells per well in 24-well plate and incubated at 2 to 5% O_2_ for 18–24 h. The cell layer was then scraped with a tool of moderate width and washed once with medium without serum and the growth medium was changed. Each antibody was treated for 24–48 h, and then cell migration was observed by microscopy at several time points. Cells migrated into the membrane were identified using a transwell of an 8.0 um pore size. Cells were seeded at the top of the transwell at a density of 2 × 10^4^ cells/100 μL in medium without serum and antibodies diluted in growth medium with 20% concentration of serum added to the bottom of the transwell. Cells were incubated for 24 h in a 37 °C incubator with 2 to 5% O_2_. After 24 h, the insert wells were washed twice with PBS, treated with 70% ethanol for 10 min, and stained with 0.2% crystal violet dye for 15 min. The captured cells in the transwell membrane were observed and counted under a microscope.

### 4.7. In Vivo Study Using Xenograft Model NRG Mice

All animal experiments were conducted in accordance with the National Research Council’s (IACUC, Gyeonggi-do, Republic of Korea) Guide to the Care and Use of Laboratory Animals. The overall procedures for in vivo experiments was reviewed and approved by the Animal Experiments Committee of Duksung Women’s University (Approval No. 2024-006-004). NRG mice (NOD-Rag2-IL2rgTm1/Rj) were purchased from Raonbio Co., Ltd. (Seoul, South Korea) and were provided by JANVIER LABS (Le Genest-Saint-isle, France). Mice were maintained in a 12 h day/12 h night environment at 23 to 27 °C with food and water provided in the absence of specific pathogens. Prior to under experiments, mice were acclimated for 1 week. Mice were divided into three groups and injected intravenously in the tail with Luc-SK-Hep1 cells (5 × 10^5^ cells/mouse) and then treated after 1 week with intraperitoneal injections of DPBS (control group) with each antibody (anti-human IL-6: 2 ng/mL; anti-human MCP1: 1 ng/mL) twice weekly. Two weeks after the start of the experiment, tumors in mice derived from cancer cells were visualized using an in vivo imaging system (VISQUE; Vieworks, Gyeonggi-do, South Korea) immediately after intraperitoneal injection with luciferin (3 mg per each mouse) once per week for 5 weeks.

### 4.8. Tissue IHC Staining Analysis

Mouse liver tissues were excised, slowly frozen in Frozen Section Compound (Leica, Hesse, Germany) under the influence of liquid nitrogen, and stored at −80 °C until the experiment. Briefly, each tissue sample was sectioned to a thickness of 5 μm and placed on a microscope slide. The tissue slices were soaked once in tap water, treated with 3% H_2_O_2_ diluted in methanol, and washed with PBS. The sections were then treated with 0.05% Triton X-100 diluted in PBS, followed by blocking buffer (10% BSA and 0.05% Tween 20 diluted in PBS) for 1 h, prior to primary antibody blotting. Primary antibodies were diluted in PBST (0.05% Tween 20 diluted in PBS) and incubated for at least 18 h at 4 °C. After incubation, the sections were washed with PBS and incubated with a secondary antibody for 2 h and a third antibody (VECTASTAIN^®^ ABC Kits (HRP), VectorLabs, Newark, CA, USA) for 1 h. The section were washed with TBS for staining and blotted with DAB substrate (VectorLabs) within 5 min. Hemotoxylin was used as a contrast stain for nucleic acid staining. The stained sections were imaged by light microscopy. Primary antibodies: HIF-1α (Novus Biologicals, Centennial, CO, USA), pSTAT3 (Cell signaling, Danvers, MA, USA), N-cadherin (Cell signaling), and MMP9 (Cell signaling).

### 4.9. NK Cytotoxicity Using LDH-Release Assay

NK cytotoxicity against target cells (Luc-SK-Hep1) was assessed by LDH release assay (CytoTox96^®^ Non-Radioactive Cytotoxicity Assay, Promega, Madison, WI, USA). Briefly, target cells were plated at a density of 5 × 10^3^ cells per well in a 96-well bottom plate at 2 to 5% O_2_ for 24 h and then co-cultured with effector cells (NK-92) at a ratio of 1:2 (T:E) for 5 h. After co-culture, the cells were centrifuged at 1600 rpm for 4 min; then, the supernatants were transferred to a new well of a fresh 96-well bottom plate and treated with CytoTox96 reagent for 30 min. Absorbance was measured at 490 nm using a microplate reader (BMG Labtech, Ortenberg, Germany).

### 4.10. Chemotaxis Assay

NK migration into the HCC microenvironment was analyzed using transwell assay. Target cells were plated at a density of 5 × 10^4^ cells per well in a 24-well plate and each incubated at 20% or 2 to 5% O_2_ for 48 h. After incubation, each well was placed in a transwell insert and the top of the 5.0 μm pore insert well was seeded with effector cells (NK-92) at a density of 4 × 10^5^ cells per well and the bottom was treated with antibodies and then co-cultured for 24 h. After co-culture, the insert wells and supernatants were removed and the cells in the bottom were carefully washed twice with DPBS, treated with 100% methanol for 5 min, and stained with 0.2% crystal violet dye for 15 min. The stained cells were imaged by microscope (×40, ×100) and dissolved in 33% acetic acid to measure absorbance at 595 nm.

### 4.11. Flow Cytometry Analysis

NK-activating receptors were analyzed using flow cytometry after the binding of fluorescent antibodies. Briefly, target cells were incubated in a 6-well plate at 2 to 5% O_2_ for 48 h and co-cultured with effector cells (NK-92) at a 1:2 (E:T) ratio for 5 h. The effector cells were then harvested and stained with anti-NKG2D-APC (eBioscience, San Diego, CA, USA), anti-NKp30-PE (BD Biosciences), NKp44-PE (BD Biosciences), and anti-CD56-APC (BD biosciences) for 30 min. After staining, the stained cells were measured by flow cytometry (Novocyte Flow Cytometer). The positive population of cells for each antibody was compensated by comparing each emission value.

### 4.12. Statistical Analysis

Data were analyzed using GraphPad Prism version 5.03 for Windows, GraphPad Software, www.graphpad.com (accessed on 9 July 2020), and Microsoft Excel. Results were presented as the mean ± SD and comparisons of means were performed by *t*-tests or by one-way ANOVA with Tukey’s multiple comparison tests. Significance between groups was indicated by a *p*-value of less than 0.05.

## 5. Conclusions

We first found high levels of MCP1 in human hepatocellular carcinoma (HCC) cell culture under hypoxic conditions. Then, blocking MCP1 with an antibody inhibited the ability of growth, migration, and invasion in human HCC cell culture under hypoxia. The effects were due to the reduction in HIF-1α and STAT3 expression in HCC under hypoxia. In addition, the blockade of MCP1 in HCC cell culture under hypoxia increased the activation of NK cells to HCC cells. These results suggest that regulation of MCP1 can be used as an indicator to induce effective NK immune surveillance in hypoxic HCC (Figure 7).

## Figures and Tables

**Figure 1 ijms-26-04900-f001:**
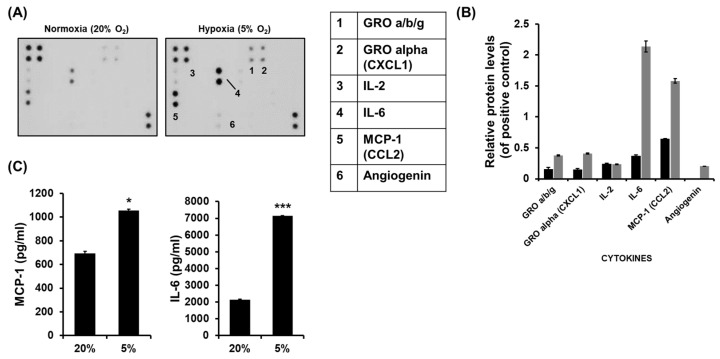
The high production of MCP1 and IL-6 by HCC cells under hypoxia. Luc-SK-Hep1 cells were seeded at a density of 5 × 10^5^ cells per well in a 6-well bottom plate and incubated for 48 h under normoxia (20% O_2_) or hypoxia (5% O_2_). After 48 h, supernatants were collected, centrifuged, and stored at −80 °C prior to experiments. Analysis of cytokines in the supernatants was performed by a cytokine array (**A**) and its quantitative graph (**B**), and by ELISA (**C**). Black and gray bars are the protein expression of normoxia and hypoxia, respectively. Means ± SD are presented from three independent experiments, and the significance between groups is set at * *p* < 0.05 and *** *p* < 0.0001.

**Figure 2 ijms-26-04900-f002:**
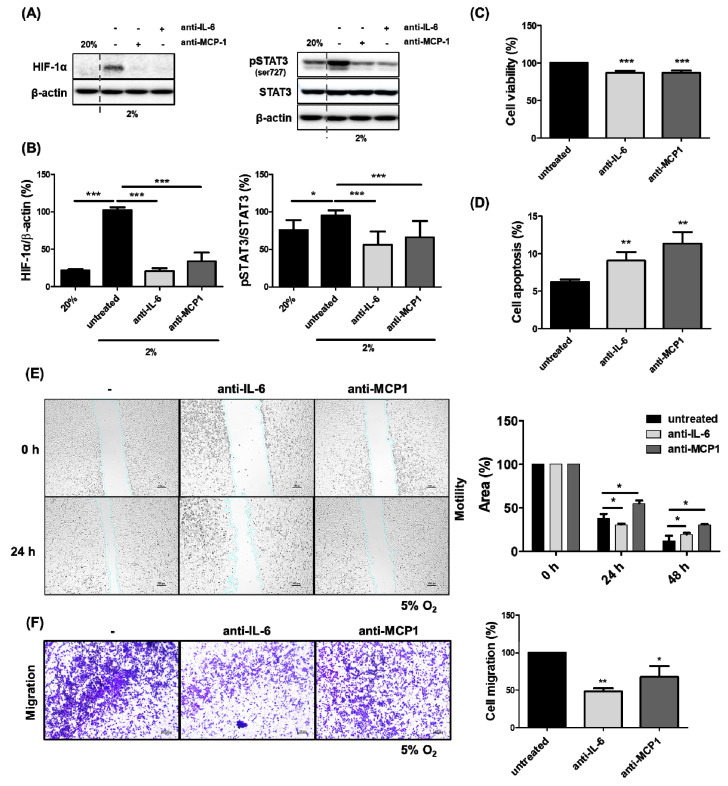
Inhibition of HCC cell growth and migration by blockade of MCP1 under hypoxia. Luc-SK-Hep1 cells were incubated with or without anti-IL-6 (2 ng/mL) or MCP1 (1 ng/mL) for 24–48 h under hypoxia (2–5% O_2_). Cell lysate was prepared with protein extraction buffer and used for Western blot analysis. Expression of HIF-1α (**A**) and (p)-STAT3 (**B**) in cells. Cell viability (**C**) and apoptosis (**D**) were analyzed by CCK-8 assay and APC Annexin V staining, respectively. Cell mobility (**E**) and migration (**F**) were analyzed by wound healing assay and transwell assay, respectively. Black, light gray and dark gray bars represent untreated, anti-IL-6 and anti-MCP1, respectively. All data represent mean ± SD after at least three independent experiments were performed and significance compared to untreated group was set at * *p* < 0.05, ** *p* < 0.001, and *** *p* < 0.0001.

**Figure 3 ijms-26-04900-f003:**
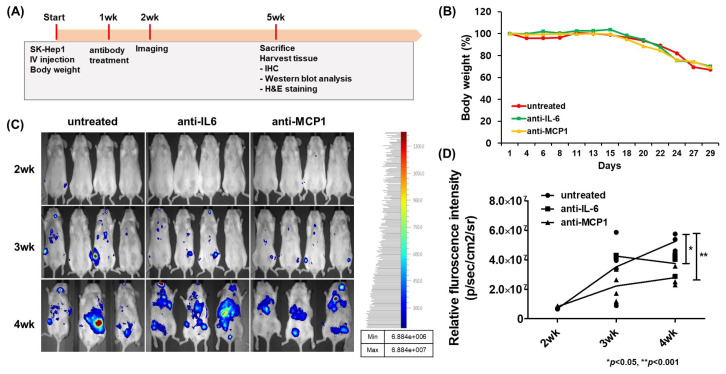
The suppression of tumor development and metastasis of HCC in mice by antibody treatment. NRG mice (5 weeks old/female) deficient in lymphocytes (T, B and NK cells) were used for in vivo experiments. The mice were injected intravenously through the tail with Luc-SK-Hep1 cells (5 × 10^6^ cells/mouse), divided into three groups, and treated after 1 week with intraperitoneal injections of DPBS, anti-IL-6 (2 μg/kg), or anti-MCP1 (1 μg/kg) twice a week until the end of this study. Tumors were visualized using an in vivo imaging system 2 weeks after the injection of the cancer cells. The overall scheme of the in vivo experiment (**A**), mouse body weight (**B**), the fluorescence intensity of luciferase in mice (**C**), and its quantitative graph (**D**) are shown. Data were analyzed by one-way ANOVA with Turkey’s multiple comparison test and significance between groups was indicated by a *p*-value of less than 0.05. * *p* < 0.05, ** *p* < 0.001.

**Figure 4 ijms-26-04900-f004:**
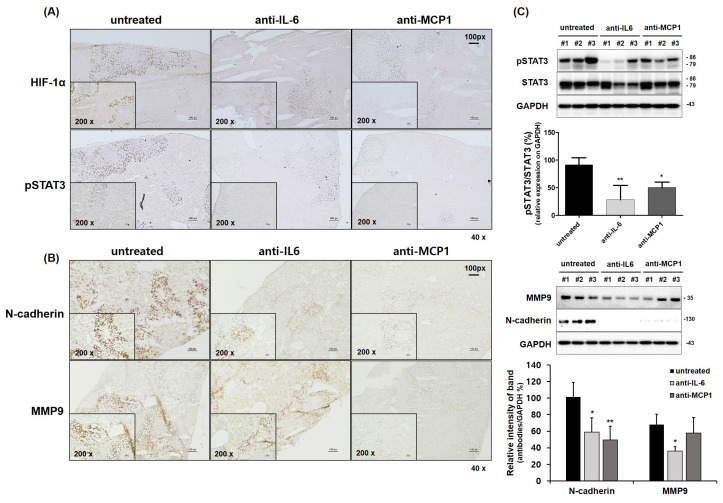
The expression of molecules associated with cancer development in mouse liver metastatic tumor tissues generated by HCC cells. Metastatic tumors in the mouse liver generated by the intravenous injection of HCC cells were partially lysed in protein extraction buffer for Western blot analysis and partially frozen in Frozen Section Compound for IHC assay. Representative images of HIF-1α, pSTAT3 (**A**), N-cadherin, and MMP9 (**B**) by IHC assay and the expression of (p)-STAT3, MMP9, and N-cadherin (**C**) by Western blot analysis in HCC metastatic liver tissue are shown. The images are in the size of 40× or 200×. Data are presented for metastatic liver tissue from at least three independent mice per group. Significance between groups was indicated by *p*-value of less than 0.05. * *p* < 0.05 and ** *p* < 0.001 vs. untreated.

**Figure 5 ijms-26-04900-f005:**
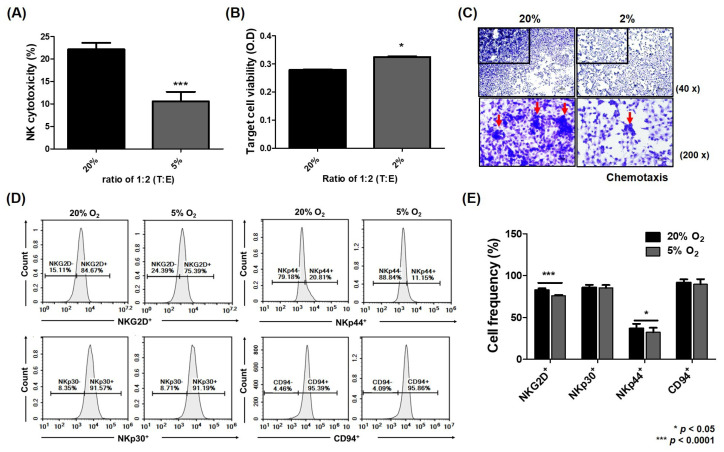
Attenuated NK function against HCC cells under hypoxia. Target cells were incubated for 48 h and co-cultured with effector cells (NK-92) at a 1:2 (T:E) ratio for 5 h. NK cytotoxicity (**A**) by LDH-release assay, target cell viability (**B**), and effector cell migration (**C**) by transwell assay, and NK activating receptors (**D**) and their quantitative graphs (**E**) by flow cytometry are shown. The red arrows indicate the NK-92 cell cluster. All data are presented as mean ± SD from three independent experiments and significance was set at *p* < 0.05. * *p* < 0.05, *** *p* < 0.0001 vs. 20% O_2_.

**Figure 6 ijms-26-04900-f006:**
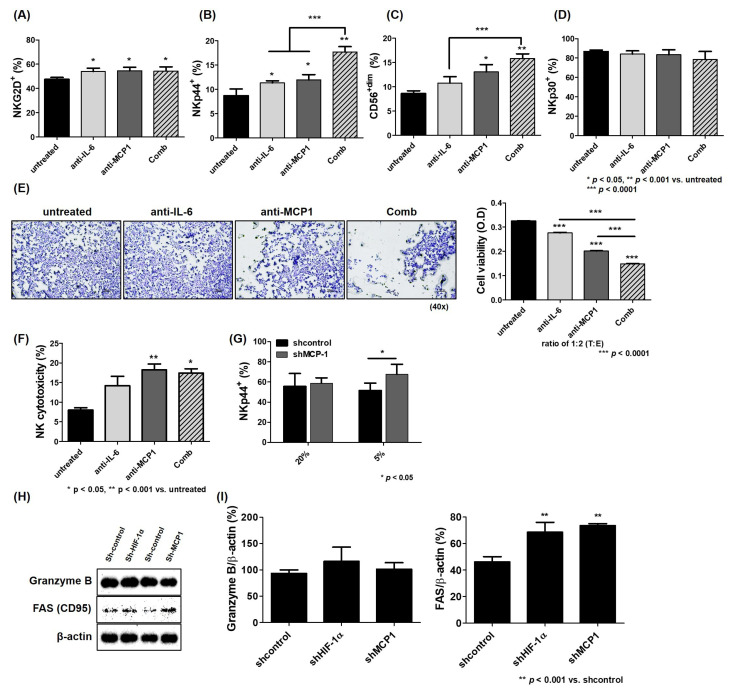
Enhancement of NK activity against HCC cells by blockade of MCP1 under hypoxia. Target cells were incubated for 48 h with human anti-IL-6 (2 ng/mL) or anti-MCP1 (1 ng/mL) and co-cultured with effector cells (NK-92) at 1:2 (T:E) ratio for 5 h. Expression of NKG2D (**A**), NKp44 (**B**), CD56^dim^ (**C**), and NKp30 (**D**) stained with respective fluorescent antibodies on surface of NK-92 cells. Target cell viability (**E**) on effector cell using transwell assay and NK cytotoxicity (**F**) by LDH-release assay. NKp44 expression (**G**) on shRNA-transfected HCC cells. Expression of granzyme B and Fas (CD95) (**H**) in co-culture of shRNA-transfected HCC cells with NK-92 cells by Western blot analysis and their quantitative graphs (**I**). All data are presented as mean ± SD in three independent experiments and significance was set at *p* < 0.05. * *p* < 0.05, ** *p* < 0.001, and *** *p* < 0.0001.

**Figure 7 ijms-26-04900-f007:**
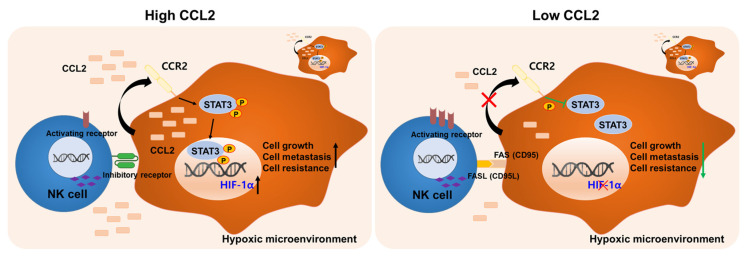
The enhancement of NK activation through the neutralization of CCL2 in the hypoxic microenvironment of HCC. The black arrow and red X represent the activation and inhibition, respectively. Green blunt arrow indicate inhibition, while ↑ represent increase and ↓ represent decrease.

## Data Availability

The original contributions presented in this study are available in the article/Appendix A. Further inquiries should be directed to the corresponding author(s).

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
