# Peer review of "Inhibition of MCP1 (CCL2) Enhances Antitumor Activity of NK Cells Against HCC Cells Under Hypoxia"

_ijms, 2025, doi:10.3390/ijms26104900_

Round 1

Reviewer 1 Report

Comments and Suggestions for Authors

The manuscript by Lee et al. reports that hypoxia induces CCL2 secretion by the HCC cell line Luc-SK-Hep1, which subsequently mediates suppressive effects on the NK cell line NK92. These findings offer novel insights into the immunosuppressive mechanisms of the tumor microenvironment (TME) and suggest MCP-1 (CCL2) inhibition as a potential therapeutic strategy.

The study employs a combination of relevant in vitro, in vivo, and molecular biology techniques. The findings are novel and of great interest to the field. However, there are several significant limitations in the experimental design that undermine the overall validity and generalizability of the conclusions:

  1. The most critical limitation is the exclusive use of a single HCC cell line (Luc-SK-Hep1) and a single NK cell line (NK92), both of which are artificial and low-fidelity models. As such, the findings may be entirely cell line–specific and cannot support definitive conclusions regarding broader biological mechanisms. I encourage authors to confirm the findings on other cell line/primary cell cultures.

  2. A mechanistic concern relates to the receptor interaction of CCL2. This chemokine primarily signals through CCR2 and CCR4. Based on my own (unpublished) data, neither the DSMZ nor the ATCC-distributed NK92 cells expresses these receptors. This discrepancy suggests that either the authors are using a variant of NK92 that expresses one of these receptors, or CCL2 may be acting through an alternative, unidentified pathway. Further investigation into the chemokine receptor profile of the NK92 cells used in this study is strongly encouraged.

Author Response

Comment 1: The most critical limitation is the exclusive use of a single HCC cell line (Luc-SK-Hep1) and a single NK cell line (NK92), both of which are artificial and low-fidelity models. As such, the findings may be entirely cell line–specific and cannot support definitive conclusions regarding broader biological mechanisms. I encourage authors to confirm the findings on other cell line/primary cell cultures.

Response 1: Thank you for your advice.
Various papers have reported the overexpression of CCL2 in solid tumors. Also, recent some studies have reported that CCL2 neutralization can induce NK cell and T cell infiltration and activation by suppressing the activity of immune-suppressive cells [Sheera R. R. et al 2025, Suguru K. et al 2021]. As mentioned in lines 286-292, CCL2 inhibition was found to reduce the activation of HIF-1α, PI3K/AKT, and STAT3, thereby influencing tumor growth and metastasis inhibition as well as the activation of immune-activating cells. However, it is not yet known whether CCL2 neutralization directly affects NK cell activation. In this study, CCL2 blockade was found to influence NK cell activation receptors and increased cytotoxic effects, suggesting the possibility that CCL2 neutralization may induce NK cell activation. In subsequent studies, we plan to conduct research on primary cell culture following CCL2 neutralization using other cell lines and animal experiments, as advised by the reviewers. Additionally, the purpose of our study is to design CAR-NK cells conjugated with a CCL2 antibody to overcome drug and effector cell resistance in cancer cells caused by CCL2 neutralization in a hypoxic microenvironment through HIF-1α expression.

Comment 2: A mechanistic concern relates to the receptor interaction of CCL2. This chemokine primarily signals through CCR2 and CCR4. Based on my own (unpublished) data, neither the DSMZ nor the ATCC-distributed NK92 cells expresses these receptors. This discrepancy suggests that either the authors are using a variant of NK92 that expresses one of these receptors, or CCL2 may be acting through an alternative, unidentified pathway. Further investigation into the chemokine receptor profile of the NK92 cells used in this study is strongly encouraged.

Response 2: Thank you for your comment. 
Recent research by Frederik F. F. and colleagues (2023, Int J Mol Sci.) showed that overexpression of CCR2 on the surface of NK-92 cells through CCR2 gene transduction results in chemotaxis and changes in receptor activity due to interaction with CCL2. However, we found that NK-92 cells do not express CCR2 in the absence of CCL2, but its expression increases in the presence of CCL2 (Supplementary Figure 3). Furthermore, these results suggest that the neutralization of CCL2 enhances NK cell activity by altering the mechanisms of cancer cells in a hypoxic microenvironment, rather than directly explaining the effects of the direct interaction between CCL2 and CCR2. This study also suggests the possibility that activating NK cells can be achieved by inhibiting the activity of HIF-1α and pSTAT3, which are expressed in cancer cells under hypoxic conditions.

Following your advice, we have revised line 457 to focus on activation rather than direct effects on chemotaxis: “In addition, blockade of MCP1 in HCC cell culture under hypoxia increased the activation of NK cells toward HCC cells.”

Reviewer 2 Report

Comments and Suggestions for Authors

Reviewer’s comments

This original article written by Lee et al. highlighted the inhibitory effects of MCP1 on the NK cell activity against HCC cells under hypoxia. This study has its own originality and seems to be very interesting. However, several data on this study were missing. Interpretation of the results obtained from this study seemed to be insufficient in several parts. Several revisions are required to improve the quality of this article. Please refer to the comments shown below.

#1. This study had in vitro and in vivo experiments and did not contain clinical trials. However, “human HCC” was selected as one of the keywords. It does not seem to be appropriate.

#2. Both 2% and 5% O2 concentrations were set as hypoxia. What was the difference between them?

#3. Sample number should be noted in each experiment.

#4. Significant differences were found in Figure 2C, 2D and 2F. Were the significances found between

untreated group and other groups, or between anti-IL-6 group and anti-MCP1 group? The authors should clearly indicate that.

#5. If the authors wanted to indicate the longitudinal changes in tumor volume (Figure 3D), a line graph was more preferable in each group.

#6. The data for western blotting in HIF-1α were missing (Figure 4C). Were the significant differences found, compared to those in untreated groups?

#7. What did red arrows indicate in Figure 5C? The data for CD94+ were missing in Figure 5E.

#8. The explanation for Figure 6 did not correspond to each figure. Figure 6A, 6B, 6C and 6D indicated the frequencies of NKG2D, NKp44, CD56dim, and NKp30, respectively. How much of HIF-1α and MCP1 genes were attenuated by shRNA plasmid transfection? The authors should indicate them.

#9. Why the cell frequency in NKp30 was not suppressed under hypoxia (Figure 5E)? Likewise, why the frequency of NKp30 was not suppressed by the treatment with anti-IL6, anti-MCP1 or the combined treatment (Figure 6D)?

#10. Why did not the combined treatment increase the frequency of NKG2D (Figure 6A)?

#11. What did the result that Granzyme activity in shHIF-1α group or shMCP1 group were approximately equivalent to that in shControl group indicate (Figure 6H)?

#12. The authors should use a schema to show the presumed mechanism by which the inhibition of MCP1 or IL6 enhanced NK cell activity against HCC cells under hypoxia.

#13. The authors should describe the data availability in this study.

Author Response

Comment 1: This study had in vitro and in vivo experiments and did not contain clinical trials. However, “human HCC” was selected as one of the keywords. It does not seem to be appropriate.

Response 1: Thanks for your comments. 
In this study, we used human hepatocellular carcinoma cell line (in vitro or in vivo experiments) and although we used a single cell line, we carefully determined that it was appropriate to focus on the effects of MCP1 (CCL2) on human hepatocellular carcinoma (HCC). Following your suggestion, we are planning a clinical trial to investigate the effects of CCL2 on the interaction between HCC and NK cells. In future studies, we plan to develop CCL2 antibody-conjugated NK cells for use in preclinical or clinical trials.

Comment 2: Both 2% and 5% O2 concentrations were set as hypoxia. What was the difference between them?

Response 2: Hypoxia in solid tumors refer to a state where oxygen concentration is between 0.5 and 2%, and hypoxia in cancer cells is confirmed by the expression of hypoxic-inducible factor (HIF)-α. In Figure 2A, HIF-1α expression can be induced after more than 24 hours at a 5% oxygen concentration; however, due to the stability of the hypoxic state at a 2% oxygen concentration, some experiments were conducted at a 2% oxygen concentration.

Comment 3: Sample number should be noted in each experiment.

Response 3: Thank you for your opinions.
We cautiously suggest that reporting sample numbers in in vitro experiments is generally not recommended. Immunohistochemistry (IHC) experiments on liver tissue in animal experiments were performed on liver tissue from each of three mice and representative images were used. this sentence is provided on lines 202-204.

Comment 4: Significant differences were found in Figure 2C, 2D and 2F. Were the significances found between untreated group and other groups, or between anti-IL-6 group and anti-MCP1 group? The authors should clearly indicate that.
Response 4: Thank you for your comments.
We have revised line 141 to read, “ significance compared to untreated group was set at *p<0.05, **p<0.001 and ***p<0.0001.” 

Comment 5: If the authors wanted to indicate the longitudinal changes in tumor volume (Figure 3D), a line graph was more preferable in each group.

Response 5: Thank you for your feedback.
We have revised a line graph in Figure 3D.

Comment 6: The data for western blotting in HIF-1α were missing (Figure 4C). Were the significant differences found, compared to those in untreated groups?

Response 6: In Figure 4C, we conducted western blot analysis for the identification of the expression of HIF-1α however, the band was not clarified (signal was weak). So, we didn’t insert the band picture of HIF-1α but as your opinion, we attached the supplementary figure.

Comment 7: What did red arrows indicate in Figure 5C? The data for CD94+ were missing in Figure 5E.

Response 7: Thank you for your advice.
We have added the data for CD94+ in Figure 5E as you suggested.

Comment 8: The explanation for Figure 6 did not correspond to each figure. Figure 6A, 6B, 6C and 6D indicated the frequencies of NKG2D, NKp44, CD56dim, and NKp30, respectively. How much of HIF-1α and MCP1 genes were attenuated by shRNA plasmid transfection? The authors should indicate them.

Response 8: Thank you for your comments.
We have revised line 216-219 to read, “Expression of NKG2D (A), NKp44 (B), CD56dim (C) and NKp30 (D) stained with respective fluo-rescent antibodies on the surface of NK-92 cells. Target cell viability (E) on effector cell using Transewell assay and NK cytotoxicity (F) by LDH-release assay.”
The results of knocking down the HIF-1α and MCP1 genes using shRNA plasmid transfection are shown in supplementary figure 1 and 2.

Comment 9: Why the cell frequency in NKp30 was not suppressed under hypoxia (Figure 5E)? Likewise, why the frequency of NKp30 was not suppressed by the treatment with anti-IL6, anti-MCP1 or the combined treatment (Figure 6D)?

Response 9:  Thank you for your comments.
The activation of NK cells is regulated by the balance between activating or inhibitory receptors. NK activating receptors are represented by NKG2D and NCRs such as NKp30, NKp44 and NKp46. However, these receptors have different signaling pathways for activating NK cells. Therefore, the lack of change in NKp30 is through to be due to differences in signaling pathways caused by ligands on the surface of cancer cells.

Comment 10: Why did not the combined treatment increase the frequency of NKG2D (Figure 6A)?

Response 10: Several studies have shown that the IL-6/JAK/Stat3 signaling pathway regulates the negative expression of NK activation receptors (NKG2D, NKp30, NKp44, etc.) (Li-Jun Xu et al., 2018; Jian Xu et al., 2019, etc.). However, the effects of CCL2 on NK activation have not been clearly reported. However, researchers speculate that CCL2 may affect NK cell function through its negative effects on activation receptors. For example, CCL2 increases the PI3K/AKT and MAPK (ERK, JNK, p38) pathways in various cells. In particular, the MAPK pathway induces the expression of inflammatory genes in cells, thereby reducing the expression of activation receptors on the surface of NK cells. Li-Jun Xu et al. (2019) demonstrated that inhibition of the IL-6/JAK/Stat3 pathway alters the expression of NKG2D ligands (MICA/B) on the surface of tumor cells. In addition, Lee et al (2019) confirmed the expression of NKG2D and NKp44 on the surface of NK-92 cells after anti-IL-6 therapy. Our results suggest that the increased expression of HIF-1α through activation of STAT3 by CCL2 and IL-6 may influence the receptor expression of natural killer cells; however, since the receptors for IL-6 and CCL2 are different, this difference in ligand expression on the surface of cancer cells may account for the effects. In addition, the ligand for NKp44 remains unidentified.

Comment 11: What did the result that Granzyme activity in shHIF-1α group or shMCP1 group were approximately equivalent to that in shControl group indicate (Figure 6H)?

Response 11: NK cytotoxicity against target cells (such as virally infected or cancer cells) is generated by two representative signaling pathways. One is the killing effect through the release of granzyme B or perforin from NK cells to cancer cells resulting in mitochondrial mediated apoptosis and another is the use of death receptor (Fas-FasL interaction) resulting in cleavage of caspase-8. Figure 6H indicates that the cytotoxic effect of NK cells is mediated by increased expression of Fas (CD95) and not by release of granzyme B.

Comment 12: The authors should use a schema to show the presumed mechanism by which the inhibition of MCP1 or IL6 enhanced NK cell activity against HCC cells under hypoxia.

Response 12: Thank you for your feedback.
The schema has been added as the figure 7 in this study at your suggestion.

Comment 13: The authors should describe the data availability in this study.
Response 13: Thank you for your comments.
We have added line 479-480 to read “Data Availability Statement”: The original contributions presented in this study are available in the article/supplementary material. Further inquiries should be directed to the corresponding author(s).”

Reviewer 3 Report

Comments and Suggestions for Authors

The authors here reported the hypoxia-dependent role of MCP1 in mediating the anti-tumor activity of NK cells against HCC cells. Through the specific inhibition of MCP1, the cytotoxicity against HCC cells is much more enhanced. The conclusion can be well supported with the provided experiment evidences. I have no other major suggestions except some minor points need to be noted.

  1. A proposed mode of action of down-regulating MCP1 in multiple pathways for the enhancement of NK cells against HCC cells can be illustrated as a figure. It will give a more clear impression to readers.
  2. The relationship between MCP1 and IL-6 is seemingly not clearly elucidated. How are they regulated in NK and HCC cells? A more discussion details can be added.
  3. In line 108,  “*p<0.0001”should be “***p<0.0001”.
  4. In Figure 6H and 6I, the β-actin images are the same. It will easily evoke the misunderstanding of reusing the same figure. The blot image of granzyme B, Fas and β-actin can be combined as one figure.

Author Response

Comment 1: A proposed mode of action of down-regulating MCP1 in multiple pathways for the enhancement of NK cells against HCC cells can be illustrated as a figure. It will give a more clear impression to readers.

Response 1: Thank you for your advice.
I have added a diagram of this study as Figure 7.

Comment 2: The relationship between MCP1 and IL-6 is seemingly not clearly elucidated. How are they regulated in NK and HCC cells? A more discussion details can be added.

Response 2: Thank you for your comments.
Several studies have reported high levels of IL-6 and MCP1 in solid tumors. In addition, our previous studies have demonstrated the effects of IL-6 on breast cancer and hepatocellular carcinoma. HIFa, which is highly expressed in hypoxia, induces cancer cell growth, metastasis and resistance. Inhibition of IL-6 and MCP-1 leads to suppression of HIF-1α expression through inhibition of STAT3 phosphorylation; however, the direct effect appears to be mediated by MCP-1. As mentioned in lines 261-265, in the treatment with anti-IL-6 and anti-MCP-1, the level of MCP-1 in the culture medium decreased when anti-IL-6 was administered. In addition, recent studies have reported that the binding site of HIF-1 is located on the promoter of MCP-1. Based on these results, although the intracellular signaling molecules are similar, the direct effect is expected to be mediated by MCP1.

In addition, as suggested by you, we have added the following content to lines 303-307 regarding the potential role of MCP1 regulation in hypoxic solid tumors in enhancing the antitumor effects of NK cells: Furthermore, IL-6 is a multifunctional cytokine that affects various physiological functions and may have systemic effects, whereas MCP1 is a chemokine with local effects. Therefore, these findings suggest that inhibition of HIF-1α by neutralizing MCP1, which is a local cytokine, may overcome hypoxia-induced resistance in solid tumors and enhance the antitumor activity of NK cells.

Comment 3: In line 108,  “*p<0.0001”should be “***p<0.0001”.

Response 3: Thank you for your point.
We have revised as per your advice.

Comment 4: In Figure 6H and 6I, the β-actin images are the same. It will easily evoke the misunderstanding of reusing the same figure. The blot image of granzyme B, Fas and β-actin can be combined as one figure.

Response 4: Thanks for your feedback.
Following your advice, we have reworked this into a combined image.

Round 2

Reviewer 2 Report

Comments and Suggestions for Authors

Round-2

Comment 1: This study had in vitro and in vivo experiments and did not contain clinical trials. However, “human HCC” was selected as one of the keywords. It does not seem to be appropriate.

Response 1: Thanks for your comments. 
In this study, we used human hepatocellular carcinoma cell line (in vitro or in vivo experiments) and although we used a single cell line, we carefully determined that it was appropriate to focus on the effects of MCP1 (CCL2) on human hepatocellular carcinoma (HCC). Following your suggestion, we are planning a clinical trial to investigate the effects of CCL2 on the interaction between HCC and NK cells. In future studies, we plan to develop CCL2 antibody-conjugated NK cells for use in preclinical or clinical trials.
→responded well

Comment 2: Both 2% and 5% O2 concentrations were set as hypoxia. What was the difference between them?

Response 2: Hypoxia in solid tumors refer to a state where oxygen concentration is between 0.5 and 2%, and hypoxia in cancer cells is confirmed by the expression of hypoxic-inducible factor (HIF)-α. In Figure 2A, HIF-1α expression can be induced after more than 24 hours at a 5% oxygen concentration; however, due to the stability of the hypoxic state at a 2% oxygen concentration, some experiments were conducted at a 2% oxygen concentration.

Why was O2 concentration was set 5% as hypoxia In Figure 1A, Figure 2E, Figure 5D, and Figure 6G?

Comment 3: Sample number should be noted in each experiment.

Response 3: Thank you for your opinions.
We cautiously suggest that reporting sample numbers in in vitro experiments is generally not recommended. Immunohistochemistry (IHC) experiments on liver tissue in animal experiments were performed on liver tissue from each of three mice and representative images were used. this sentence is provided on lines 202-204.

→The sentence was not found on lines 202-204.

Comment 4: Significant differences were found in Figure 2C, 2D and 2F. Were the significances found between untreated group and other groups, or between anti-IL-6 group and anti-MCP1 group? The authors should clearly indicate that.
Response 4: Thank you for your comments.
We have revised line 141 to read, “significance compared to untreated group was set at *p<0.05, **p<0.001 and ***p<0.0001.” 

 →responded well.

Comment 5: If the authors wanted to indicate the longitudinal changes in tumor volume (Figure 3D), a line graph was more preferable in each group.

Response 5: Thank you for your feedback.
We have revised a line graph in Figure 3D.

→responded well.

Comment 6: The data for western blotting in HIF-1α were missing (Figure 4C). Were the significant differences found, compared to those in untreated groups?

Response 6: In Figure 4C, we conducted western blot analysis for the identification of the expression of HIF-1α however, the band was not clarified (signal was weak). So, we didn’t insert the band picture of HIF-1α but as your opinion, we attached the supplementary figure.

→responded well.

Comment 7: What did red arrows indicate in Figure 5C? The data for CD94+ were missing in Figure 5E.

Response 7: Thank you for your advice.
We have added the data for CD94+ in Figure 5E as you suggested.

The data for CD94+ were found. However, what did red arrows indicate in Figure 5C?

Comment 8: The explanation for Figure 6 did not correspond to each figure. Figure 6A, 6B, 6C and 6D indicated the frequencies of NKG2D, NKp44, CD56dim, and NKp30, respectively. How much of HIF-1α and MCP1 genes were attenuated by shRNA plasmid transfection? The authors should indicate them.

Response 8: Thank you for your comments.
We have revised line 216-219 to read, “Expression of NKG2D (A), NKp44 (B), CD56dim (C) and NKp30 (D) stained with respective fluo-rescent antibodies on the surface of NK-92 cells. Target cell viability (E) on effector cell using Transewell assay and NK cytotoxicity (F) by LDH-release assay.”
The results of knocking down the HIF-1α and MCP1 genes using shRNA plasmid transfection are shown in supplementary figure 1 and 2.

→responded well.

Comment 9: Why the cell frequency in NKp30 was not suppressed under hypoxia (Figure 5E)? Likewise, why the frequency of NKp30 was not suppressed by the treatment with anti-IL6, anti-MCP1 or the combined treatment (Figure 6D)?

Response 9:  Thank you for your comments.
The activation of NK cells is regulated by the balance between activating or inhibitory receptors. NK activating receptors are represented by NKG2D and NCRs such as NKp30, NKp44 and NKp46. However, these receptors have different signaling pathways for activating NK cells. Therefore, the lack of change in NKp30 is through to be due to differences in signaling pathways caused by ligands on the surface of cancer cells.

Authors responded very well to the comment. However, the reason why the cell frequency in NKp30 was not suppressed under hypoxia should be described in “Discussion”. Likewise, the reason why the frequency of NKp30 was not suppressed by the treatment with anti-IL6, anti-MCP1 or the combined treatment should be also mentioned in “Discussion”

Comment 10: Why did not the combined treatment increase the frequency of NKG2D (Figure 6A)?

Response 10: Several studies have shown that the IL-6/JAK/Stat3 signaling pathway regulates the negative expression of NK activation receptors (NKG2D, NKp30, NKp44, etc.) (Li-Jun Xu et al., 2018; Jian Xu et al., 2019, etc.). However, the effects of CCL2 on NK activation have not been clearly reported. However, researchers speculate that CCL2 may affect NK cell function through its negative effects on activation receptors. For example, CCL2 increases the PI3K/AKT and MAPK (ERK, JNK, p38) pathways in various cells. In particular, the MAPK pathway induces the expression of inflammatory genes in cells, thereby reducing the expression of activation receptors on the surface of NK cells. Li-Jun Xu et al. (2019) demonstrated that inhibition of the IL-6/JAK/Stat3 pathway alters the expression of NKG2D ligands (MICA/B) on the surface of tumor cells. In addition, Lee et al (2019) confirmed the expression of NKG2D and NKp44 on the surface of NK-92 cells after anti-IL-6 therapy. Our results suggest that the increased expression of HIF-1α through activation of STAT3 by CCL2 and IL-6 may influence the receptor expression of natural killer cells; however, since the receptors for IL-6 and CCL2 are different, this difference in ligand expression on the surface of cancer cells may account for the effects. In addition, the ligand for NKp44 remains unidentified.

→The authors responded well to the comment. The reason why the combined treatment did not increase the frequency of NKG2D should be described in “Discussion”.

Comment 11: What did the result that Granzyme activity in shHIF-1α group or shMCP1 group were approximately equivalent to that in shControl group indicate (Figure 6H)?

Response 11: NK cytotoxicity against target cells (such as virally infected or cancer cells) is generated by two representative signaling pathways. One is the killing effect through the release of granzyme B or perforin from NK cells to cancer cells resulting in mitochondrial mediated apoptosis and another is the use of death receptor (Fas-FasL interaction) resulting in cleavage of caspase-8. Figure 6H indicates that the cytotoxic effect of NK cells is mediated by increased expression of Fas (CD95) and not by release of granzyme B.

→The authors should describe a role of granzyme B.

Comment 12: The authors should use a schema to show the presumed mechanism by which the inhibition of MCP1 or IL6 enhanced NK cell activity against HCC cells under hypoxia.

Response 12: Thank you for your feedback.
The schema has been added as the figure 7 in this study at your suggestion.

→responded well.

Comment 13: The authors should describe the data availability in this study.
Response 13: Thank you for your comments.
We have added line 479-480 to read “Data Availability Statement”: The original contributions presented in this study are available in the article/supplementary material. Further inquiries should be directed to the corresponding author(s).”

→responded well.

Author Response

Comment 2: Why was O2 concentration was set 5% as hypoxia In Figure 1A, Figure 2E, Figure 5D, and Figure 6G?

Response 2: We apologize for any confusion this may have caused. First, we performed a measurement of cytokines at 5% or 20% oxygen concentrations. The 5% oxygen concentration is a reference to a previous study. The comment has been added to lines 320-321 with the reference to read: "To induce a hypoxic environment, we used the oxygen concentration settings mentioned in previous studies [59]. Therefore, most of the experiments investigating the effects of antibodies on HCC cells were performed under the first condition. However, protein expression was only performed under 2% oxygen conditions for stability (Figure 2A and B).

Comment 3: The sentence was not found on lines 202-204.

Response 3: We sincerely apologize for our mistake. We were not aware that the relevant section had been changed during the revision of the paper. The content on line 204-205 was written as follows: "Data are presented for metastatic liver tissue from at least three independent mice per group.

Comment 7: The data for CD94+ were found. However, what did red arrows indicate in Figure 5C?

Response 7: Thank you for your comments.
The red arrow is the NK-92 cell cluster. Content was added to line 211-212 as follows: "The red arrow indicates the NK-92 cell cluster."

Comment 9-10: Authors responded very well to the comment. However, the reason why the cell frequency in NKp30 was not suppressed under hypoxia should be described in “Discussion”. Likewise, the reason why the frequency of NKp30 was not suppressed by the treatment with anti-IL6, anti-MCP1 or the combined treatment should be also mentioned in “Discussion”

The authors responded well to the comment. The reason why the combined treatment did not increase the frequency of NKG2D should be described in “Discussion”.

Response 9-10: Thank you for your advice.
We have added the content in the discussion (lines 296-307) as your advice.

Comment 11: The authors should describe a role of granzyme B.

Response 11: Thank you for your feedback.
We have described the role of granzyme B in Results (lines 193-197). NK cells can kill the target cells (such as virus-infected or transformed cells) by inducing apoptosis of the target cells. apoptosis is generated by two mechanisms. One is the release of granzyme B or perforin from effector (NK) cells into target cells, or another is the death receptor signals (Fas-FasL interaction).

Reviewer 3 Report

Comments and Suggestions for Authors

The resubmitted manuscript has been well revised according to the suggestions. It can be accepted for publication in its present form.

Author Response

Thank you for your interest in our manuscript.
Your feedback is greatly appreciated.

Round 3

Reviewer 2 Report

Comments and Suggestions for Authors

Round-3

Comment 2: Why was O2 concentration was set 5% as hypoxia In Figure 1A, Figure 2E, Figure 5D, and Figure 6G?

Response 2: We apologize for any confusion this may have caused. First, we performed a measurement of cytokines at 5% or 20% oxygen concentrations. The 5% oxygen concentration is a reference to a previous study. The comment has been added to lines 320-321 with the reference to read: "To induce a hypoxic environment, we used the oxygen concentration settings mentioned in previous studies [59]. Therefore, most of the experiments investigating the effects of antibodies on HCC cells were performed under the first condition. However, protein expression was only performed under 2% oxygen conditions for stability (Figure 2A and B).

The authors responded well. The reasons for 2% O2 and 5%O2 as hypoxia should be clearly mentioned in Materials and Methods (1.1. Cell maintenance).

Comment 3: The sentence was not found on lines 202-204.

Response 3: We sincerely apologize for our mistake. We were not aware that the relevant section had been changed during the revision of the paper. The content on line 204-205 was written as follows: "Data are presented for metastatic liver tissue from at least three independent mice per group.

→Responded well.

Comment 7: The data for CD94+ were found. However, what did red arrows indicate in Figure 5C?

Response 7: Thank you for your comments.
The red arrow is the NK-92 cell cluster. Content was added to line 211-212 as follows: "The red arrow indicates the NK-92 cell cluster."

→Responded well.

Comment 9-10: Authors responded very well to the comment. However, the reason why the cell frequency in NKp30 was not suppressed under hypoxia should be described in “Discussion”. Likewise, the reason why the frequency of NKp30 was not suppressed by the treatment with anti-IL6, anti-MCP1 or the combined treatment should be also mentioned in “Discussion”

The authors responded well to the comment. The reason why the combined treatment did not increase the frequency of NKG2D should be described in “Discussion”.

Response 9-10: Thank you for your advice.
We have added the content in the discussion (lines 296-307) as your advice.

→Responded well.

Comment 11: The authors should describe a role of granzyme B.

Response 11: Thank you for your feedback.
We have described the role of granzyme B in Results (lines 193-197). NK cells can kill the target cells (such as virus-infected or transformed cells) by inducing apoptosis of the target cells. apoptosis is generated by two mechanisms. One is the release of granzyme B or perforin from effector (NK) cells into target cells, or another is the death receptor signals (Fas-FasL interaction).

→Responded well.

Author Response

Comment 2: The authors responded well. The reasons for 2% O2 and 5%O2 as hypoxia should be clearly mentioned in Materials and Methods (1.1. Cell maintenance).

Response 2: At your suggestion, we have revised lines 326 to 328 (part of Materials and Methods) to read like this:: "The hypoxic condition was 5% oxygen for most experiments, but 2% oxygen for the protein expression experiment."